# Polish Adaptation and Psychometric Properties of the Long- and Short-Form Interfaith Spirituality Scale

**DOI:** 10.3390/ijerph192013274

**Published:** 2022-10-14

**Authors:** Janusz Surzykiewicz, Sebastian Binyamin Skalski, Małgorzata Niesiobędzka, Loren L. Toussaint, Karol Konaszewski

**Affiliations:** 1Faculty of Philosophy and Education, Catholic University of Eichstätt-Ingolstadt, 85072 Eichstätt, Germany; 2Faculty of Education, Cardinal Stefan Wyszyński University in Warsaw, 01938 Warsaw, Poland; 3Faculty of Education, University of Bialystok, 15328 Bialystok, Poland; 4Department of Psychology, Luther College, Decorah, IA 52101, USA

**Keywords:** interfaith spirituality scale, polish adaptation, IFS, spirituality scale

## Abstract

Spirituality is widely believed to play an important role in securing health, and modern health care is increasingly being combined with spiritual care. This state of affairs is generating widespread interest in the construct from practitioners and researchers, resulting in the need to develop accurate and comprehensive measures of spirituality. The purpose of this study was to adapt the Polish version of the Interfaith Spirituality Scale (IFS), which consists of twenty-two statements, as well as its short version, including four statements. The IFS is not limited to any one religious denomination, making it possible to conduct research across diverse communities. The analyses were conducted on a sample of 642 Poles aged 18–68, 48% of whom were women. The Polish version of the scale showed high internal consistency (α = 0.96 for the IFS and α = 0.81 for the short version). Confirmatory factor analysis showed that the structure of the IFS consists of a unitary second-order factor with four first-order factors (direct connection with the creator, asceticism, meditation, and divine love). On the other hand, the structure of the short version is unifactorial. There were positive correlations of the IFS with another measure of spirituality, post-traumatic growth severity, mental well-being, and ecological behaviour, as well as negative correlations with post-traumatic stress disorder symptom severity and depressiveness; these confirmed the high validity of the tool. The results suggest that the IFS may be useful in the study of spirituality on Polish soil.

## 1. Introduction

The vast majority of people in the world declare themselves to be religious or spiritual [1]. Although religiosity is closely related to spirituality for religious people, primarily due to its frequent (albeit not always) reference to the non-empirical realm and consideration of transcendent reality, researchers have consistently pointed to the existence of differences underlying these related but nonetheless distinct constructs (e.g., [2]). Religiosity manifests itself in external and institutional rituals of devotion or organised worship, while spirituality focuses more on the inner state of being and places a stronger emphasis on the individual and personal, as well as subjectively signifying the existential [3,4]. Such a subtle distinction is particularly important in scholarly discourse on the potential existence of a moderating relationship between spirituality and religion, since a person can simultaneously be religious and spiritual in their own way [5]. This indistinct overlap between religion and spirituality also makes it difficult to develop a precise and comprehensive measure of spirituality.

In addition to the association of spirituality with religiosity, past research has revealed that this variable has a significant impact on psychophysical health. Reports provide a range of evidence for the negative association of spirituality with psychopathology, depression, generalised anxiety, existential anxiety, and the severity of post-traumatic stress disorder (PTSD) [6,7,8,9,10]. On the other hand, authors have consistently highlighted positive associations of spirituality with mental and spiritual well-being, life satisfaction, resilience, positive affect, ecological behaviour, gratitude and awe, optimism or post-traumatic growth (PTG) [4,11,12,13,14,15]. In a study by Kira et al. [16], spirituality was also identified as being an effective strategy for coping with existential threats during the COVID-19 pandemic, as well as a more effective method for coping with PTSD symptoms compared to other available methods such as social support, resilience, or the will to exist. The psychological benefits of spirituality have resulted in modern health care increasingly being combined with spiritual care. According to Puchalski [17], seriously ill patients value the spiritual experiences they undertake with their doctors. Patients place a high value on being with their doctor, being close to them, and being able to talk about issues that are relevant and important to them. In such a situation, one of the key challenges facing doctors is to help people find meaning and acceptance in suffering and chronic illness. A similar approach is taken by the Association of American Medical Colleges (AAMC), which has issued recommendations to health care practitioners, suggesting that they view health as a process through which individuals can maintain a sense of coherence and meaning in life in the face of changes within themselves, such as illness [18,19]. Thus, spirituality should be seen as a factor that contributes to the health of many people.

The modern understanding of spirituality must first be grasped from the point of view of its ancient roots. In its original etymology, it is derived from the Greek *pneuma*, popularised by the Latin expression *spiritus*, which expresses the peculiar principle of life, its force, expressing (as opposed to corporeality) the immaterial dimension of human existence, understood in terms of spirit and soul. This principle is externalised in the psychosomatic aspects of breathing, ‘sensory’ experiences, vitality, and also in cognitive-emotional abilities such as self-reflection, self-awareness, and the ability to search for and make meaning [20,21]. Hence, in general, spirituality describes a person’s relationship with the inner world, their deeper dimension of existence, which is formless, but also transcendent [22,23,24,25,26,27]. From a philosophical and anthropological point of view, spirituality is special in its expression of the uniqueness of man revealed in our capacity for spiritual experiences, as well as our need for them. Spirituality also represents an important and integral dimension of human psychic experiences [28].

Most conceptualisations define spirituality as a multidimensional phenomenon. Elkins [29] proposed that spirituality encompasses six essential characteristics: spirituality is universal; it is a human phenomenon; its core is phenomenological; it is our ability to respond to the sacred; it is characterised by ‘mystical energy’; and its ultimate goal is compassion. In such a perception, spirituality has become a phenomenon that refers to the totality of life, which is associated with a number of psychic qualities. Emmons [30] described spirituality as a deep sense of belonging and openness to the infinite; as a psychic phenomenon relating to the highest truths; and as a pathway to experiencing transcendence in a broad sense. A characteristic feature of spirituality emphasised by most authors is its connection with the sacred [31,32,33]. A popular definition of spirituality proposed by Hill and Pargament [34] describes the phenomenon as the “search for the sacred” (p. 65), with ‘sacred’ referring to persons and objects of ultimate truth and devotion. Furthermore, Hill and Pargament [34] argued that the “polarization of religion and spirituality into institutional and individual domains ignores the fact that all forms of spiritual expression unfold in a social context” (p. 64), and they deem spirituality to be a broader construct that can potentially be expressed through religious or other contextual channels.

Today, the common denominator of many, though not all, conceptions of spirituality is its reference to transcendence [35,36,37]. To measure this phenomenon, the most common scales used are those developed by Catlin et al. [38], Grant et al. [39], Harper and Schulte-Murray [40], and Howard et al. [41], although these are generally limited to specific religious beliefs, making them inapplicable to different populations. In response to this problem, Kira et al. [42] proposed the concept of the interfaith spirituality (IFS) based on the personal narrative and heuristics used to make sense of a person’s time-limited existence, as well as to cope with existence-threatening events. Such heuristics are created by people who believe in the existence of a sacred power, regardless of their different belief systems. The aforementioned heuristics are based on beliefs about the fundamental nature of human existence and involve a belief in a sacred higher power and the ability to self-transcend, i.e., to transcend oneself and be open to the world, namely the people one can approach and the senses it can fulfil [5]. Self-transcendence is “a constitutive characteristic of being human that [is] always point [ed], and is directed, to something other than itself. Actually, being human profoundly means to be open to the world, a world, that is, which is replete with other beings to encounter and with meanings to fulfil” [43] (p. 97). Following Conn [44], Sperry [45] identifies self-transcendence as a relational construct whereby an individual transcends self-preoccupation and becomes intimate with another. In this intimacy with the other, the self is known. Self-transcendence, then, is a dialectic process between independence and intimacy. That is, the self is known only through relations with others.

The IFS contains four basic spiritual aspects that are universal to various religions: (1) *Direct connection with the creator*, (2) *Asceticism*, (3) *Divine love*, and (4) *Meditation*. Together, these cluster into a second-order global spirituality factor [46]. The IFS has been found to be related to alleviation of existential turmoil, anxiety, and related spiritual struggles [42]. The structure of IFS has been shown to be invariant across religions. To date, analyses have been conducted on samples of Christians, Muslims, Jews, atheists, and agnostics. The IFS has also been shown to be invariant across national groups, including Egyptians, Kuwaiti Turks, Syrians/Palestinians, and Britons [46]. In light of these data, the IFS appears to be an interesting tool for measuring spirituality that can be applied to different populations, regardless of their belief system.

Since most spirituality questionnaires available in Poland are limited by references to specific religious denominations (especially Catholicism; e.g., [47]), which prevents their use in heterogeneous communities, the purpose of this study was to adapt and preliminarily assess the psychometric properties of the Polish language long and short version of the IFS, including its validity and reliability.

## 2. Materials and Methods

### 2.1. Participants and Procedure

The study was conducted with the approval of the Ethics Committee of the Institute of Psychology of the Polish Academy of Sciences in Warsaw (No. 14/05/2021). The study sample comprised 642 Poles aged 18–68 (*M* = 34.51, *SD* = 8.86), 48% of whom were women. Of the participants, 72% declared a Catholic religious affiliation, 5% Protestant, 3% Orthodox, 1% Muslim, 1% Jehovah’s Witnesses, while 18% declared no religious affiliation. Participation in the survey was preceded by informed consent and did not involve any recruitment criteria. The survey was conducted in May 2022 via the Prolific survey panel (data were collected in Google Forms and then exported to an aggregate spreadsheet excluding data that would identify participants). The survey procedure consisted of completing two questionnaires to measure spirituality, specifically the IFS and the Scale of Spirituality [47] (this scale was only administered to the Catholic population due to the fact that some statements were specific to Catholicism), mental well-being, effects of trauma (PTSD and PTG), ecological behaviour, and depression symptoms. The average time to complete the questionnaires was 15 min. Participants were paid GBP 2.50 for taking part in the survey.

### 2.2. Measures

In this study, we conducted a Polish adaptation of the IFS [46] to self-describe spirituality in an interfaith paradigm, understood as sensing the existence of a direct relationship with one’s creator and the ability to transcend oneself. With ‘creator’ meaning the force that brought everything into existence, further defined according to individual perception. The IFS translation was carried out by two independent translators, and then compared and analysed by a specialist in philosophical terminology in a subsequent step. The original version of IFS was translated into Polish by three independent translators with a high proficiency in English. The translations were adjusted to the final version of the scale by the authors of the present study. Next, the final version was back-translated into English by three independent translators with a high level of proficiency in Polish. Any differences between the original and back-translated version of the scale were discussed, amended, and accepted by the authors of the study. The translation of the scale was carried out in accordance with accepted principles developed for the purposes of intercultural research [48], based on the original English version. The IFS consists of 22 statements (see Appendix A) arranged into four factors: (1) *Direct connection with the creator*, which refers to feeling the presence of the creator (God or a higher being) in our lives, as well as having a relationship with them, (2) *Asceticism*, which describes self-discipline and virtue, such as limiting one’s love for material things in favour of developing spirituality, (3) *Divine love*, which describes perceiving the creator’s love for us and expressing our love for them, and (4) *Meditation*, which refers to perceiving nature to provoke reflection, as well as seeking knowledge regarding human existence. These four factors are combined in the second-order factor. In addition, Kira et al. [46] proposed a short version of the scale, which consists of four statements arranged into a single factor called ‘spirituality’. In both versions of the scale, the subject expresses their attitude toward each statement on a four-point Likert scale, where 1 = “*Disagree*” and 4 = “*Agree*”.

In addition, for the purpose of assessing the validity of the IFS, we used the Scale of Spirituality by Skowronski and Bartoszewski [47] to measure spirituality, understood as the search for universal truth that enables one to discover one’s own meaning and significance in the world. This Polish scale consists of 36 statements arranged into six factors: (1) *Religious spirituality* (*α* = 0.96), which is related to maintaining a connection with a spiritual guru; (2) *Spirituality as an expanding consciousness* (*α* = 0.83), which describes an attempt to learn (understand) oneself and others; (3) *Spirituality as searching for meaning* (*α* = 0.86), which refers to the search for answers to existential questions; (4) *Spirituality as sensitivity to art* (*α* = 0.85), describing spiritual experiences while participating in cultural events; (5) *Spirituality as doing good* (*α* = 0.83), which refers to caring for loved ones; and (6) *Sensitivity to inner beauty* (*α* = 0.85), which assesses the validity of moral life choices and connection to nature. The participant expresses their attitude toward each statement on a four-point Likert scale, where 1 = “*Definitely not*” and 4 = “*Definitely yes*”. Example statements include: “*I believe in the communion of saints*” and “*I am sensitive to the injustice of other people*”.

The World Health Organization’s 5-item Well-being Index (WHO-5) [49] in Polish [50] was used to measure mental well-being. The WHO-5 is a unidimensional questionnaire measuring positive well-being synonyms to mental health during the last 14 days. It consists of five statements arranged into a single factor (*α* = 0.89). The participant expresses their attitude toward each statement on a six-point Likert scale: 0 = “*None of the time*” a 5 = “*All of the time*”. Sample items include: “*I have felt calm and relaxed”* and “*I have felt active and vigorous”*.

The Short Form of the Changes in Outlook Questionnaire (SF-CiOQ) (Joseph et al., 2006) in Polish standardisation [51] was used to assess the degree of PTG and PTSD symptoms. The questionnaire includes ten statements arranged into two factors: *Positive effects* (*α* = 0.85) and *Negative effects* (*α* = 0.83). The participant expresses their attitude toward each statement on a six-point Likert scale, where 1 = “*Strongly Disagree*” and 6 = “*Strongly Agree*”. Sample items include: “*I no longer take people or things for granted”* and “*I have very little trust in other people now”*.

The Pro-environmental Behavior Scale [52] was used to measure ecological behaviour. The tool includes 16 statements arranged into a single factor (α = 0.79) that identify behaviours related to recycling, energy and water conservation, shopping, and moving around. In the Polish version [15], the participant expresses their attitude toward each statement on a five-point Likert scale, where 1 = “*Definitely not*” and 5 = “*Definitely yes*”. Sample items include: “*Buying seasonal fruits/regional vegetables”* and “*No car in household”.*

The Patient Health Questionnaire-9 (PHQ-9) developed by Kroenke et al. [53] in Polish [54] was used to assess depression symptoms. This single-factor scale (*α* = 0.88) consists of nine questions about depression symptoms derived from DSM-IV diagnostic criteria. The participant expresses their attitude toward each statement (according to the frequency of the symptom over the past two weeks) on a four-point Likert scale, where 0 = “*Not at all*” and 3 = “*Almost every day*”. Sample items include: “*Feeling down, depressed, or hopeless”* and “*Trouble falling or staying asleep, or sleeping too much”*.

### 2.3. Statistical Analyses

Statistical analysis of the data was carried out using IBM SPSS Statistics 27 and IBM SPSS Amos 27 programmes. The data obtained were normally distributed, so parametric tests could be used (verified by the Kolmogorov–Smirnov test). Confirmatory factor analysis (CFA) was used to assess the structure of the IFS. We used the following goodness of fit indices: the goodness of fit index (GFI); the adjusted GFI (AGFI); the confirmatory fit index (CFI)-required value: ratio > 0.90; the root mean square error of approximation (RMSEA)-required value: ratio < 0.08; and the chi-square (*X*^2^)/degrees of freedom (*df*)-required value: ratio < 2 and statistically insignificant *X*^2^ test [55]. The reliability of the IFS was calculated using the Cronbach’s α and the McDonald’s ω coefficients. A generally accepted rule of thumb is that an α and ω of 0.60–0.70 indicates an acceptable level of reliability. The content validity ratio (CVR) calculated using Lawshe’s [56] method was used to assess content relevance. Pearson’s r correlation analysis was used to determine the relationships between the variables. Additionally, analysis of variance (ANOVA) was used to determine differences. The Gabriel test was used for post hoc comparisons; the significance level was set at *p* ≤ 0.050.

## 3. Results

The means obtained in the study are illustrated in Table 1. All items that were part of a given IFS factor and the short version of the scale presented satisfactory discriminatory power, i.e., correlated with its total score (when excluding the item in question from the scale) at a level above 0.50.

The content validity of the Polish version of the IFS was assessed by competent judges (five psychologists) according to Lawshe’s [56] method. The CVR for each statement exceeded the required value of 0.75.

Theoretical validity was assessed using CFA using the maximum likelihood method. A model including a second-order factor with four first-order factors (as in the original version, see [46]) obtained goodness-of-fit indices: X^2^ (205) = 237.80, *p* = 0.058; X^2^/df = 1.16; GFI = 0.966; AGFI = 0.966; RMSEA = 0.064 (0.059,0.069;90% CI); CFI = 0.966 (see Figure 1). For the short version of the IFS, the values of the goodness-of-fit indices allowed acceptance of its univariate structure: X^2^ (2) = 1.90, *p* = 0.387; X^2^/df = 0.95; GFI = 0.999; AGFI = 0.995; RMSEA = 0.001 (0.000, 0.007; 90% CI); CFI = 1.00 (see Figure 2).

Internal consistency was assessed using Cronbach’s Alpha coefficient and McDonald’s ω. The Cronbach’s Alpha coefficient for the IFS scale was 0.96, *α* = 0.97 for *Direct connection with the creating force*, *α* = 0.91 for *Asceticism*, *α* = 0.83 for *Meditation*, *α* = 0.83 for *Divine love*, and *α* = 0.81 for the short version of the scale. In addition, the value of McDonald’s *ω* coefficient was analysed, which was 0.96 for IFS, *ω* = 0.97 for *Direct connection with the creating force*, *ω* = 0.91 for *Asceticism*, *ω* = 0.82 for *Meditation*, *ω* = 0.83 for *Divine love*, and *ω* = 0.80 for the short version of the scale.

Convergent validity was estimated using correlation analysis. The IFS had a strong positive correlation with subscales of the Scale of Spirituality [47]. In addition, we observed moderate positive correlations between IFS and mental well-being and PTG, and negative correlations between IFS and PTSD symptoms and depressiveness. IFS also had a weak positive correlation with ecological behaviour. Detailed values of correlation coefficients with IFS component factors are presented in Table 1; the table also shows values of intercorrelation coefficients within IFS.

Finally, we assessed the effect of sociodemographic variables on Polish IFS scores. Age was found to be associated with higher IFS scores (*r* = 0.19, *p* < 0.001), *Direct connection with the creator* (*r* = 0.17, *p* < 0.001), *Asceticism* (*r* = 0.17, *p* < 0.001), *Divine love* (*r* = 0.13, *p* = 0.002), *Meditation* (*r* = 0.18, *p* < 0.001), and the short version of the scale (*r* = 0.17, *p* < 0.001). Non-believers showed lower spirituality scores on the full version of the scale (*F*_(1,640)_ = 149.09, *p* < 0.001; *M*_believers_ = 47.61, *SD* = 16.71; *M*_non-believers_ = 34.34, *SD* = 9.94), *Direct connection with the creator* (*F*_(1,640)_ = 219.34, *p* < 0.001; *M*_believers_ = 15.80, *SD* = 7.26; *M*_non-believers_ = 9.25, *SD* = 3.14), *Asceticism* (*F*_(1,640)_ = 46.40, *p* < 0.001; *M*_believers_ = 11.88, *SD* = 4.03; *M*_non-believers_ = 9.64, *SD* = 4.28), *Divine love* (*F*_(1,640)_ = 161.10, *p* < 0.001; *M*_believers_ = 5.94, *SD* = 2.66; *M*_non-believers_ = 4.88, *SD* = 2.36), *Meditation* (*F*_(1,640)_ = 50.87, *p* < 0.001; *M*_believers_ = 13.98, *SD* = 4.67; *M*_non-believers_ = 11.62, *SD* = 3.62), and in the short form of the IFS (*F*_(1,640)_ = 125.08, *p* < 0.001; *M*_believers_ = 8.95, *SD* = 3.17; *M*_non-believers_ = 6.58, *SD* = 2.06). Type of religion and sex did not differentiate the results in a statistically significant way.

## 4. Discussion

Since the time of Ancient Greece, educators believed that the health of any individual had a solid spiritual basis [57]. For Hippocrates, spirituality was “the nature of healing, that is, the vital force-pneuma (or spirit)-that God gives to man” [58], while ‘healing’ itself can be defined as the sense of well-being that derives from an intensified awareness of wholeness and integration of all dimensions of our being [59]. The critical role of spirituality in securing health and supporting healing processes has also been confirmed by numerous meta-analyses [60,61,62]. Moreover, modern health care is increasingly being combined with spiritual care [17]. This state of affairs is generating widespread interest in the construct among practitioners and researchers, and, as a result, is generating the need to develop accurate and comprehensive measures of spirituality. Unfortunately, most of the existing tools have been limited to specific belief systems and cannot be applied to broader communities [5]. In such a situation, we decided to conduct a Polish adaptation of the IFS, which is one of the first scales for assessing spirituality in an interfaith paradigm [46]. In validation studies, the Polish version of the IFS and its shortened version showed very good psychometric properties, and the scales met the basic reliability requirements. The structure of the IFS turned out to be the same as the original version of the tool, which means that in the Polish population, we reproduced the factor structure that has been obtained so far in studies on samples of Egyptians, Kuwaiti Turks, Syrians/Palestinians, and Britons [46].

Strong positive correlations of the IFS with the factors of another Polish spirituality scale [47], as well as medium correlations with mental well-being, PTG, and weak correlations with ecological behaviour, indicate the tool’s high convergent validity. Spirituality is believed to influence well-being and PTG through two main pathways, namely behavioural self-regulation (e.g., avoiding stimulants and engaging in activities to adapt to the event) and emotional self-regulation (e.g., providing hope and love) [63]. Numerous studies have also ascertained that spirituality reduces stress, thereby contributing to reduced cardiovascular reactivity, hypothalamic–pituitary–adrenal axis activation, and inflammation, which may consequently improve health outcomes [64]. In the case of ecological behaviour, Skalski et al. [15] noted that spirituality encourages the promotion of ecology through central educational and pro-social values. In the adaptation study, we also showed negative mean associations of spirituality with PTSD symptom severity and depressive symptoms, which also demonstrates the accuracy of the convergent Polish version of the IFS. The consensus in the literature is that spirituality plays a protective role in the context of trauma, because as a multidimensional construct involving a range of intrapersonal and interpersonal social aspects, it can reduce the severity of PTSD symptoms and depression among individuals [65].

Age has been shown to be positively related to IFS scores. A deepening of faith with ageing has been widely observed in the literature, and spirituality has even been identified as a developmental factor in the ageing process [66]. In addition, studies have consistently shown a relationship between age and the use of religious beliefs or activities to cope with stress [67,68]. Although the exact mechanism is yet to be demonstrated, it is speculated that religion and spirituality may provide a worldview in which suffering and death appear to be better understood and accepted. Alternatively, they may shape self-esteem in a more resilient way than other sources that decline at a commensurate rate with declining health and/or increasing age [69].

Non-believers showed lower scores on all dimensions of the IFS (as well as the shortened version of the scale) compared to believers (religion alone did not significantly differentiate the results). This appears to be understandable since, spirituality and religion are strongly related [1]. Although some people equate spirituality with religious activity and even use the words interchangeably [70], spirituality and religiosity are separate, albeit overlapping, constructs [71]. For example, Saucier et al. [72] demonstrated that subjective spirituality and tradition-oriented religiosity are empirically highly independent of each other in a large sample of American adults, confirming the divergence between the two constructs. In our study, intergroup differences between believers and non-believers on meditation and asceticism were significantly smaller than the other IFS dimensions. Kira et al. [42] observed that while belief in a sacred power is absent in non-believers, the heuristics of self-transcendence may be basic in the so-called secular spirituality and beliefs of non-believers, which may explain the results obtained.

This psychometric study has limitations. First of all, the vast majority of participants were Catholics (it is widely estimated that the population of Poland is composed of approximately 80% Catholics [18]), which made it impossible to conduct more detailed analyses on the impact of religion on the scale structure. However, on the basis of the statistical analyses conducted, the assessment of competent judges in terms of content relevance, and the previous literature, we put forth that the IFS is suitable for use with groups with different belief systems. However, further research in this area and invariance analysis are required to generalise the findings. Second, in the study, we did not assess the temporal stability of the scale. We did, however, make the assumption that spirituality is not a fixed phenomenon and can change under the influence of experiences and interventions.

## 5. Conclusions

The IFS is Poland’s first tool to assess spirituality in an interfaith paradigm. In the validation study we conducted, the scale showed very good psychometric properties. We would suggest, based on our findings, that the IFS can thus be used by Polish researchers and practitioners. In addition, as a result of its formulation, an abbreviated version of the IFS can be used with patients with attention deficits and in broader research projects where there is a need for extensive data collection. Our results also have implications that point to the crucial role of spirituality in securing health. Given our results, it seems that interventions promoting interfaith spirituality can reduce the negative effects of trauma and enhance mental health and quality of life.

## Figures and Tables

**Figure 1 ijerph-19-13274-f001:**
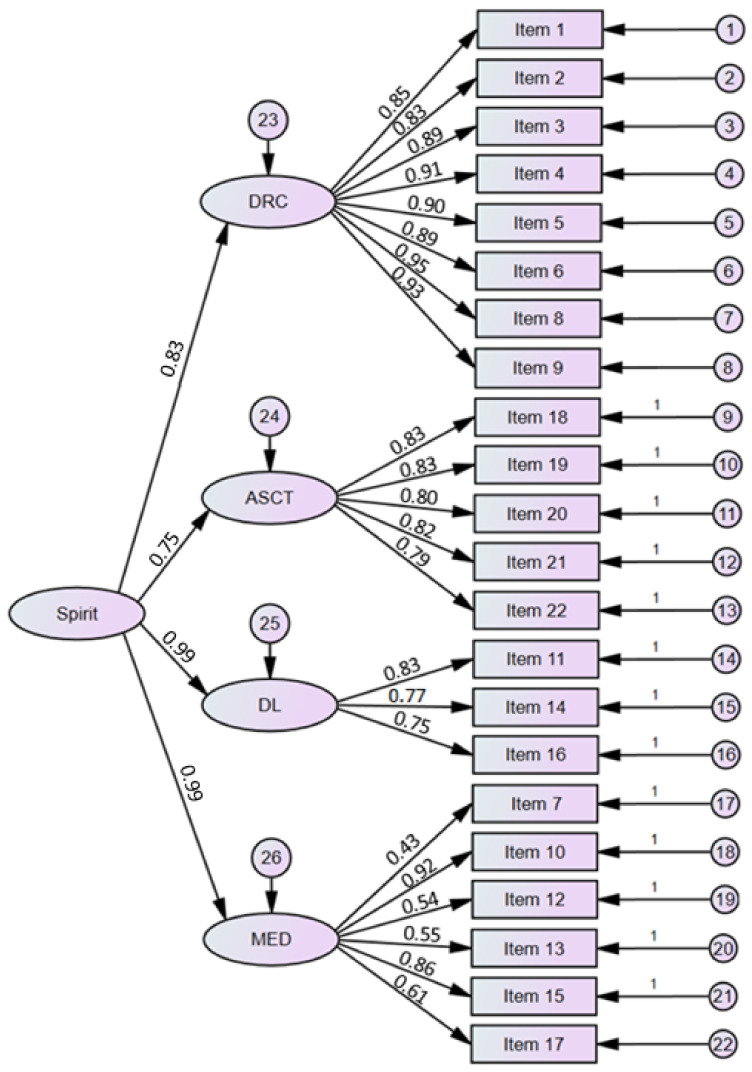
Structure of the Polish version of the Interfaith Spirituality Scale (DRC = direct connection with the creator, ASCT = asceticism, DL = divine love, and MED = meditation).

**Figure 2 ijerph-19-13274-f002:**
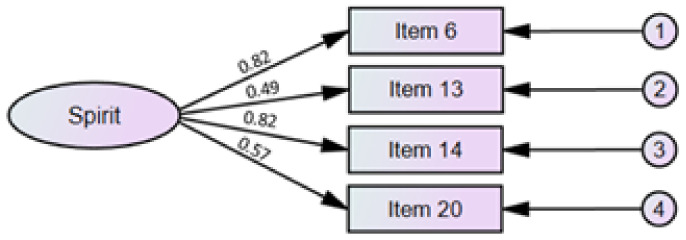
Structure of the short version of the Interfaith Spirituality Scale.

**Table 1 ijerph-19-13274-t001:** Descriptive statistics and correlations.

	M (SD)	IFS	DRC	ASCT	DL	MED	IFS-4
Religious spirituality ^a^	21.80 (10.33)	0.80 ***	0.86 ***	0.52 ***	0.58 ***	0.78 ***	0.73 ***
Spirituality as expanding consciousness ^a^	13.75 (2.08)	0.36 ***	0.21 ***	0.38 ***	0.44 ***	0.23 ***	0.35 ***
Spirituality as searching for meaning ^a^	13.71 (3.87)	0.40 ***	0.28 ***	0.38 ***	0.59 ***	0.26 ***	0.46 ***
Spirituality as sensitivity to art ^a^	10.47 (3.18)	0.37 ***	0.18 ***	0.45 ***	0.48 ***	0.22 ***	0.36 ***
Spirituality as doing good ^a^	15.89 (2.83)	0.39 ***	0.28 ***	0.38 ***	0.39 ***	0.33 ***	0.36 ***
Sensitivity to inner beauty ^a^	20.34 (2.72)	0.32 ***	0.21 ***	0.32 ***	0.41 ***	0.19 ***	0.31 ***
Mental well-being	13.95 (5.24)	0.25 ***	0.28 ***	0.26 ***	0.24 ***	0.27 ***	0.24 ***
PTG	14. 64 (4.33)	0.17 ***	0.19 ***	0.22 ***	0.18 ***	0.22 ***	0.18 ***
Ecological behaviour	56.38 (10.69)	0.08 *	0.08 *	0.16 ***	0.15 ***	0.09 *	0.08 *
PTSD	11.59 (5.62)	–0.20 ***	–0.27 ***	–0.22 ***	–0.24 ***	–0.20 ***	–0.21 ***
Depressiveness	9.98 (5.79)	–0.23 ***	–0.23 ***	–0.23 ***	–0.22 ***	–0.20 ***	–0.23 ***
Interfaith spirituality (IFS)	41.00 (15.27)	-					
Direct connection with the creator (DRC)	12.54 (6.49)	0.89 ***	-				
Asceticism (ASCT)	10.77 (4.31)	0.82 ***	0.55 ***	-			
Divine love (DL)	4.89 (2.36)	0.89 ***	0.86 ***	0.62 ***	-		
Meditation (MED)	12.81 (4.35)	0.88 ***	0.65 ***	0.74 ***	0.69 ***	-	
Short form of the IFS (IFS-4)	7.77 (2.92)	0.94 ***	0.79 ***	0.81 ***	0.87 ***	0.83 ***	-

^a^ The analysis was only conducted on Catholics (*N* = 462); *** *p* < 0.001, * *p* < 0.05.

## Data Availability

The data presented in this study are available on request from the corresponding author agreed to the published version of the manuscript.

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
