# Peer review of "Polish Adaptation and Psychometric Properties of the Long- and Short-Form Interfaith Spirituality Scale"

_ijerph, 2022, doi:10.3390/ijerph192013274_

Round 1
Reviewer 1 Report
This is a crucial paper about interfaith spirituality. I think that the IFS will be good support for spirituality research in Poland. Congratulations to the authors. I only have some minor suggestions for the authors to consider before printing the article:
Please add a period to the end of the sentence "On the other hand, the structure of the short version is unifactorial" in the abstract.
I would like to know if gender differentiated the results of your scale.
In the last paragraph of the discussion, you write, "it is widely estimated that the population of Poland is comprised of approximately 80% Catholics". Could you provide a citation for these statistics?
Author Response
This is a crucial paper about interfaith spirituality. I think that the IFS will be good support for spirituality research in Poland. Congratulations to the authors. I only have some minor suggestions for the authors to consider before printing the article:
Please add a period to the end of the sentence "On the other hand, the structure of the short version is unifactorial" in the abstract.
We added a period to the end of this sentence in the abstract.
I would like to know if gender differentiated the results of your scale.
Please note that there is already a sentence on this subject in our text, “Type of religion and sex did not differentiate the results in a statistically significant way” (p. 7).
In the last paragraph of the discussion, you write, "it is widely estimated that the population of Poland is comprised of approximately 80% Catholics". Could you provide a citation for these statistics?
We have added the citation.
Reviewer 2 Report
Appropriate background information and literature survey evidence has been provided. Methodology and procedure have been clearly presented and the results appropriately discussed with reference to various theories.
Lines 68,69. Please check journal style guidelines on whether pneuma and spiritus (non-English words) should be italicized.
Is it possible to use a different word from "tainted" or "contaminated" (e.g. Lines 99, 123 etc). For instance, the sentence may be rephrased to something like "these are influenced by Christian beliefs" or "influenced by and are relevant only to specific religions and therefore inapplicable to religions in general"
Lines 164-165. Probably there is a typo at "used:" and the colon is unnecessary, the lines may be combined.
Author Response
Appropriate background information and literature survey evidence has been provided. Methodology and procedure have been clearly presented and the results appropriately discussed with reference to various theories.
Lines 68,69. Please check journal style guidelines on whether pneuma and spiritus (non-English words) should be italicized.
As recommended, we have added the italicized notation.
Is it possible to use a different word from "tainted" or "contaminated" (e.g. Lines 99, 123 etc). For instance, the sentence may be rephrased to something like "these are influenced by Christian beliefs" or "influenced by and are relevant only to specific religions and therefore inapplicable to religions in general".
We have now replaced the terms “tainted” or “contaminated” with another word “limited” on lines 18, 100, 128, and 294.
Lines 164-165. Probably there is a typo at "used:" and the colon is unnecessary, the lines may be combined.
According to your comment, we removed the colon after “used” and combined the lines (p. 4).
Reviewer 3 Report
This is a fascinating article. I appreciate the authors wanting to validate a measure of spirituality for the Polish people. It is needed.
Two concerns: (1) much more time is needed describing the translation process. There is a citation and brief mention of translation in the results section. The entire article depends on how this measure was translated into Polish. (2) Self-transcendence is an important spiritual concept, and more space needs to be devoted to it. I would recommend (a) Sperry, L. (2012). Spirituality in clinical practice (2nd ed.). Routledge. and (b) Conn, W. E. (1998). The desiring self: Rooting pastoral counseling and spiritual direction in self transcendence. Paulist Press. This idea of self-transcendence is important for the ways in which IFS operationalizes spirituality.
Addressing these two issues would improve this article immensely.
Author Response
(1) much more time is needed describing the translation process. There is a citation and brief mention of translation in the results section. The entire article depends on how this measure was translated into Polish.
We added: “The IFS translation was carried out by two independent translators, and then compared and analysed by a specialist in philosophical terminology in a subsequent step. The original version of IFS was translated into Polish by three independent translators with a high proficiency in English. The translations were adjusted to the final version of the scale by the authors of the present study. Next, the final version was back-translated into English by three independent translators with a high level of proficiency in Polish. Any differences between the original and back-translated version of the scale were discussed, amended, and accepted by the authors of the study. The translation of the scale was carried out in accordance with accepted principles developed for the purposes of intercultural research [46], based on the original English version.” (p 4.).
Self-transcendence is an important spiritual concept, and more space needs to be devoted to it. I would recommend (a) Sperry, L. (2012). Spirituality in clinical practice (2nd ed.). Routledge. and (b) Conn, W. E. (1998). The desiring self: Rooting pastoral counseling and spiritual direction in self transcendence. Paulist Press. This idea of self-transcendence is important for the ways in which IFS operationalizes spirituality.
We added: „Following Conn [44], Sperry [45] identifies self-transcendence as a relational construct whereby an individual transcends self-preoccupation and becomes intimate with another. In this intimacy with the other, the self is known. Self-transcendence, then, is a dialectic process between independence and intimacy. That is, the self is known only through relations with others” (p. 3).
Round 2
Reviewer 3 Report
Authors addressed my concerns in the revision process.